# A Whole Education Approach to Inclusive Education: An Integrated Model to Guide Planning, Policy, and Provision

Neil Kenny [1,2,*], Selina McCoy [3] and James O'Higgins Norman [2]

1 School of Inclusive and Special Education, Institute of Education, Dublin City University, Dublin 9, D09 W6Y4 Dublin, Ireland
2 DCU Anti-Bullying Centre, Dublin City University, D09 N920 Dublin, Ireland
3 Economic and Social Research Institute, Whitaker Square, Sir John Rogerson's Quay, Dublin 2, D02 K138 Dublin, Ireland; selina.mccoy@esri.ie
* Correspondence: neil.kenny@dcu.ie

**Abstract:** Inclusion is an important aspect of achieving the Sustainable Development Goals (SDGs) in education. This article will discuss the significance of recent developments in the field of school-based bullying prevention and evaluate their applicability to the policy discourse of inclusive education. Both inclusive education and school-based bullying remain contested phenomena without a clear consensus regarding their definition or how to operationalise them as concepts within school policy or practice effectively. UNESCO's Scientific Committee has recently proposed the Whole Education Approach, which conceptualises a holistic, socially engaged, and interconnected vision for policy development in addressing school-based bullying prevention. Importantly, the Whole Education Approach conceptualises incidences of bullying as indicative of a deficit of care and support within the surrounding social environment, thus adopting an ecological and relational focus regarding bullying prevention. In addition, bullying prevention is viewed as requiring coherent collaboration between the school, family, and other relevant stakeholders in the local social community surrounding the school. This includes government funding, resource policies and national legal frameworks. This article argues that this approach may also have utility within inclusive education policy in supporting a more integrated and holistic promotion of social inclusion, underpinning equal opportunities in recognition of the diverse needs of all learners in schools. This article discusses the details of the Whole Education Approach and emphasises how this framework can also address educational inclusion by adopting an integrated, multi-elemental focus on supporting collaboration across stakeholders relevant to the lives of pupils within schools.

**Keywords:** inclusive education; school-based bullying prevention; whole education approach; education policy; additional educational needs

## 1. Introduction

In recent decades, international policy frameworks, such as the Convention on the Rights of the Child [1], have heavily influenced education policy development, moving towards a greater emphasis on inclusive provision within mainstream schools. This article aims to discuss the relevance of recent developments in the field of school-based bullying intervention and assess their applicability to the policy discourse of inclusive education [2,3]. Internationally, the issue of children's and young people's well-being in school settings is becoming increasingly important, with the United Nations recently proposing that a safe and inclusive schooling environment is directly related to societal well-being [3]. A crucial aspect of achieving the Sustainable Development Goals (SDGs) is ensuring equitable and inclusive education for all and addressing issues like bullying in schools. Educational inclusion, in a similar manner to school-based bullying, has become a central priority of educational policy in recent decades [4]. However, it is noteworthy that both

inclusive education and school-based bullying remain contested phenomena without a clear consensus regarding their definition or how to operationalise them as concepts within school policy or practice effectively. This article will discuss the proposals recently developed by the UNESCO scientific committee [2] to address this tension within both policy and practice related to school-based bullying prevention and suggest why such perspectives may also be relevant in addressing challenges within research and policy in inclusive education.

This article will comprise two sections, with the first providing an overview of the current literature regarding school-based bullying prevention and inclusive education provision. The second section will provide a detailed description of the Whole Education Approach proposed by the UNESCO scientific committee [2] and will propose its relevance to the domain of inclusive education policy and provision.

*1:. A Whole Education Approach to School-Based bullying prevention*

The phenomenon of school-based bullying has been much researched in recent decades but the literature exploring approaches to prevention or school-based intervention remains a contested domain [2,5]. According to Olweus [6] (p. 770), "being bullied by peers represents a serious violation of the fundamental rights of the child or youth exposed" and management of this phenomenon is a priority for the provision of appropriate education. However, Olweus's definition of bullying is increasingly critiqued and contested [2,7,8] and the evidence base for interventions to manage incidences of bullying within schools is mixed [3,9,10]. In more recent years, the proliferation of the internet and smartphones among adolescents has led to the inclusion of cyberbullying or online bullying within the literature, adding a further layer of complexity and nuance to the field [3,10]. Given the role of the internet and cyberbullying in relations between children and young people in schools, incidences of bullying between pupils can take place outside of school premises, further complicating the role of schools in managing experiences of bullying between pupils.

There has been a traditional focus on social-ecological approaches to bullying prevention programmes in education [11]. However, consistent evidence has emerged to suggest weak and varied outcomes for interventions based on a whole-school model, raising concerns regarding the effectiveness of whole-school intervention programmes. Researchers found that most anti-bullying programmes only reduce school bullying perpetration and victimisation by 19–20% and 15–16%, respectively, according to Gaffney et al. [8]. While whole-school approaches have been recommended in policy and practice for several decades, the UNESCO Scientific Committee suggests that a whole-school approach may be ineffective as it puts too much responsibility on single schools to tackle bullying from within their own resources [2]. Such a narrow focus would not give sufficient recognition to the important role played by the wider education system, local community attitudes, and those of wider society [2]. Schools exist within a wider education system and society, which acknowledges the role factors external to a school play in the management of relational concepts such as school-based bullying [2]. The panel of experts consulted by UNESCO highlighted the need to develop an approach that was broader than a single whole-school approach. Consequently, a wider Whole Education Approach was proposed by the UNESCO Scientific Committee, which gives a broader role for policy, funding, and wider social factors in how school-based bullying can be addressed more effectively [2,3].

## 1.1. Educational Policy Movement towards Inclusive Provision

The recent developments in theory and policy regarding school-based bullying prevention resonate with the field of inclusive education, which has developed in importance and focus in recent decades [12,13]. Inspired by social justice ideas such as the Convention on the Rights of the Child [1] and the Salamanca statement [14], many European countries have developed policies and implemented practices to promote inclusive education [15,16]. Since 1992, many definitions of inclusive education have been developed, and efforts to change special education have been undertaken. The United Nations Convention on the Rights of Persons with Disabilities recognises the right of every child to an inclusive

education, leading to inclusive education becoming a legal and moral imperative [17]. UNESCO's "Global Education Monitoring Report 2020: Inclusion and education: All Means All" report [4] emphasised the need for a comprehensive framework for whole school inclusion to provide a holistic and high-quality education to all children and young people. This viewpoint emphasises that education for all is the foundation of inclusion in education. However, it is noteworthy that there remains no consensus on a definition or clear understanding of inclusive education as a concept [13]. There is also disagreement regarding how educational inclusion should be operationalised to guide practice within schools [13,16]. This has led researchers and policy analysts to explore how wider funding, policy, and social factors interact with the work of inclusive provision within schools at both national [18] and international levels [14]. The lack of consensus may relate to the roots of educational provision for students with disabilities and additional learning needs (AENs), which had traditionally taken the form of segregated provision within adapted special education settings [14,18]. Such previous models for educational provision for divergent pupil populations focused on appropriate educational provision within separate school or class settings [13,19].

The origins of inclusive education are rooted in special education research that questioned the efficacy of separate special education classes in the 1960s. These older segregated/specialised models of educational provision for students with AENs were based on the argued need for specialised and highly adapted pedagogical approaches and settings to provide the highest quality education [16]. According to these perspectives, which were often informed by the needs of pupils with complex profiles, education was best delivered by specifically trained staff in an adapted, segregated environment [13]. The priority was to provide adapted education designed to meet the needs of those particular pupils with disabilities, or AEN, and that setting was seen to best support their educational participation. The segregated nature of the provision meant that mainstream education remained unimpacted and unchanged, and pupils with AEN were often not afforded opportunities to participate with their age-matched peers [19]. Concerns about segregated education, however, noted the overrepresentation of students from minority groups in special education provision and the stigma of labelling and civil rights issues. In other words, while arguments for the appropriateness of adapted, segregated, and specialised provision focused on the individual profile of the needs of pupils, actual practices within schools showed intersecting influences from wider societal and demographic factors [13,20,21]. Such concerns were also augmented by considerations regarding long-term impacts for pupils along their full life-span trajectories and how they could participate as citizens in society in the future. These tensions led to a paradigm shift [22] in theorising and policy for the education of learners with special or additional educational needs in the 1970s and 1980s, with a widespread movement away from segregated "special" education provision and an increasing emphasis on a more inclusive model [13,18].

A long-term understanding of social inclusion became more influential, which viewed child development as being heavily influenced by social values, access to educational institutions, and technological innovations [23]. Social participation and inclusion require that children be given opportunities to participate in a shared learning environment where they can develop a sense of belonging as members of the community and grow with their peers [24]. In short, shared inclusive educational provision within mainstream school settings was recommended to support long-term social inclusion within society [15]. As such, policy moved from special education being provided in a segregated setting to emphasising inclusion in a shared educational environment [13,18]. This shifted focus towards inclusion in schools, requiring cohesion and collaboration across staff in an inclusive school environment, and away from an emphasis on segregated sites for the education of pupils with AEN.

*1.2. Inclusion as a Place*

Given the laudable aims of this educational policy focus, it is perhaps concerning that there remains no consensus on a definition or clear understanding of inclusive education as a concept [13]. There are a range of divergent perspectives, from the idea that special education is itself a form of inclusive education to the viewpoint that inclusion should specify a form of education provision where all pupils are learning together in the same inclusive classroom [25,26]. For example, person-centred approaches that focused on mainstream class-based provision for learners with AEN argued for human difference to be celebrated as a resource rather than an educational challenge [14,27]. An alternative perspective emphasised the need to change school structures or practices to be more inclusive, thus reducing the focus on the individual profiles of particular pupils [28]. Indeed, Ainscow et al. [29] argued that inclusive education should become a feature of school improvement processes, thus emphasising the need to consider the role of the whole school community in developing inclusive cultures and practices within the whole organisation. This perspective was echoed by Clarke, Dyson, and Milward [28], who defined inclusion as "extending the scope of ordinary schools so they can include a greater diversity of children". A perhaps unintended consequence of linking inclusive education practices to school improvement policy has been that inclusion has become linked to the school as a place [15]. The linking of school improvement and inclusion as a place within schools maintained the pre-existing tension between designated separate special education provision and mainstream inclusion [13]. The focus moved to the relocation of special education practices within mainstream classrooms, which could be argued to be a form of integration rather than inclusion [14,29,30].

This lack of agreed definitions and specificity regarding how inclusion within education is to be enacted has led to significant diversity in approach and practice within schools internationally and nationally in some circumstances [14,18]. It has also led to a patchy and somewhat mixed level of "buy-in" from schools and mainstream teachers, who perceive inclusion as a cause of practical difficulties rather than a benefit to students in their classrooms [31]. Teacher attitudes towards educational inclusion and the implementation of inclusive practices have been a subject of debate and complexity [32]. This has resulted in ambivalent attitudes among school staff and a wide diversity regarding how schools interpret inclusive provision, particularly concerning more complex learning needs [33]. Moreover, the allocation of resources and provision in special educational needs (SEN) practice has been marked by inequalities [34].

Consequently, there is an increased demand for specialised support and funding due to the growing number of students identified with AENs in mainstream classes or in designated special classes attached to mainstream schools [35]. However, there have been reported delays and refusals in accessing assessments or in the provision of para-educational supports to schools to support access to appropriate educational provision [35]. Overall, the field of inclusive education faces challenges in terms of teacher attitudes, resource allocation, and the government's commitment to inclusive practices [18]. The increasing number of children with AEN and the disproportionate attendance of some cohorts of the pupil population in special classes or segregated provision raise questions about the accessibility and effectiveness of inclusive education [36].

Policy initiatives developed to provide guidance for inclusive practice in schools, such as the Inclusive Education Framework [36] or the Embracing Diversity Toolkit [37], outline multi-layered, complex models for inclusion [20,38]. As such, these frameworks require a change to existing practices or structures within schools, which can be perceived as challenging for teachers or school leaders. This can lead to schools domesticating policy initiatives [39] by choosing their own interpretation such that it aligns with existing preferences or systems within individual schools [14,40].

Given these circumstances, some have noted that inclusive education policy and practice are currently at a crossroads, with the path forward being unclear (Shevlin and Banks, 2021). The holistic and comprehensive vision for inclusion in mainstream education

is also not always clearly represented as intended within policy initiatives [14]. For example, the provision in Ireland for autistic pupils has seen a 400% increase in the number of separate special classes designated for autistic pupils. While these classes are attached to mainstream schools and operate within national inclusive education policies [41], they often serve as a form of segregated and separate provision [14,19,34]. In other words, this approach centres choices for educational provision based on diagnostic characteristics of the particular pupil, with an increasingly prevalent choice being segregation based on an autistic diagnosis. The recent inspectorate report warned of the reintroduction of segregation by accident [13,34]. The persistence of segregation via the provision of designated classes, particularly for autistic pupils, may function as a quickly implementable administrative convenience for the provision of specialised school places [13,35]. However, while this may be a temporary policy response to the frustrations of families seeking appropriate educational options, there remains a lack of policy linkage to guide coherent education policy development, funding models, and the provision of appropriate long-term placements and social inclusion for pupils with AEN in the long term [13,35].

For coherent inclusion within education, there will need to be collaborative agreement on what the aims of inclusive practice are and how they can be achieved [32]. This will involve the participation and agreement of all within the school community. This approach highlights the limitations of what can be achieved by a whole school perspective regarding inclusion and foregrounds discussions regarding the need for a Whole Education Approach put forward by the UNESCO Scientific Committee. The latter embeds an integrated perspective regarding inclusive education, which encompasses a wider societal vision that considers values or attitudes across the social fabric that surrounds schools and government resource policies that fund inclusive educational practices enacted within schools [2].

## 2. A Whole Education Approach to Inclusion

UNESCO's scientific committee has recently attempted to revise the approach to school-based bullying towards a perspective they have identified as a Whole Education Approach [2]. This approach situates the school within the wider social context in which it exists, inclusive of the wider education community (and within society more broadly), and considers the technologies that support relationships in this broader conceptualisation. This approach takes a broader perspective regarding bullying and is heavily influenced by a deeper understanding of the school context and social-ecological theory [11] than has been considered previously [3]. This perspective moves away from viewing bullying as an individual behavioural choice of students whose role remains consistent across their school careers [8]. Evidence has shown that roles within bullying experiences change over time as social influences, norms, and expectations vary [42], affirming a view of bullying as a relational phenomenon influenced by social-ecological systems surrounding individuals [2]. For example, roles observed in bullying experiences among students were seen to change across the school transition from primary to post-primary schools [43], across classes in the same school [42], and across different age ranges [44].

An alternative definition defines school bullying as in-person and online behaviour between students within a social network that causes physical, emotional, or social harm to targeted students [2]. It is characterised by an imbalance of power that is enabled or inhibited by the social and institutional norms and context of schools and the education system. This perspective argues that school bullying implies an absence of effective responses and care towards the target by peers and adults [3]. Based on the consultation with the panel of experts in a study by O'Higgins Norman et al., [3], nine core components of a whole-education approach were proposed, which will now be outlined (see Figure 1. Below). These have significant relevance to inclusive education, and the following sections will discuss each component, which will also be discussed in the context of inclusive provision in schools.

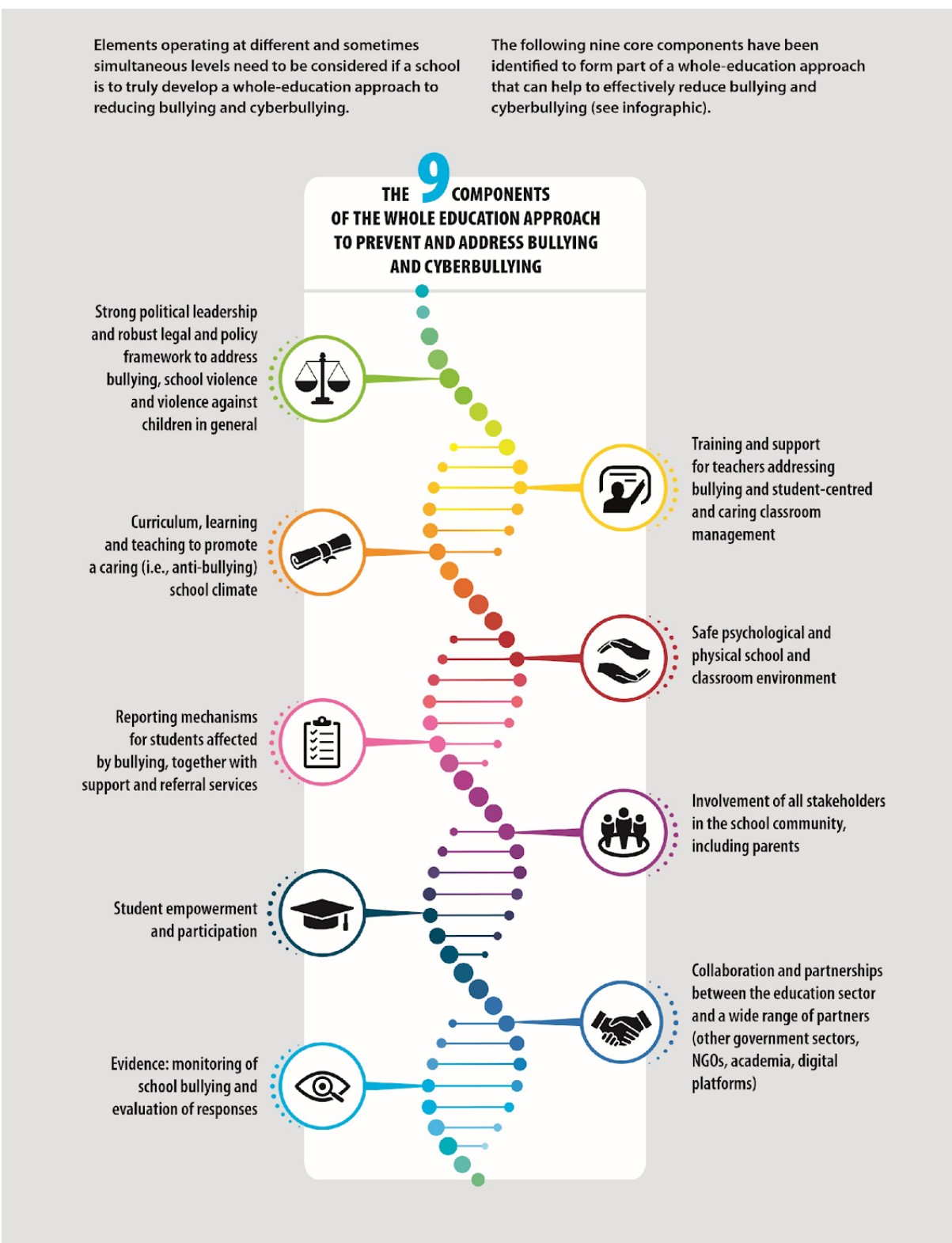

**Figure 1.** UNESCO Whole Education Approach to bullying [45].

*2.1. Strong Political Leadership*

The need to have strong political leadership across government, society, and also at the school level was found to be a core component of a comprehensive whole-education approach to tackling bullying. It has been argued [46] that insufficient attention has

been given to leadership as a key factor in addressing school-based bullying, leading to ineffective or disparate approaches to address the phenomenon [8].

This foregrounds the important role of national anti-bullying laws, policies, frameworks, and guidelines from the government to inform or support the development of school or community policies/procedures. Such laws and national guidelines should also address/include online or internet-based cyberbullying as part of a holistic and comprehensive perspective on school-based bullying prevention. The extent to which political leadership prioritises tackling school bullying will have a definite impact on the confidence of local school leaders to implement anti-bullying initiatives [2]. In 2022, the Cineáltas: Action Plan on Bullying was published by the Irish Government to update the national approach to addressing bullying in Ireland. This action plan aligns with the four key areas of the Wellbeing Policy Statement and Framework for Practice. It also adopts UNESCO's Whole Education Approach regarding school-based bullying [4].

Clear leadership in the provision of both legal and policy visions for inclusive education is also key [13,18]. However, the existing ecology of inclusive education policy and modes of provision at both the national and international levels is characterised by extensive diversity and a lack of coherence [18]. While there is a clear focus in policy on special class provision for autistic pupils across the Irish education system, there remains a lacuna regarding how such classes should operate within schools or how such classes can be utilised to support inclusive practice across the school more generally [13]. The Irish Government is scheduled to publish procedures for schools for the implementation of the Action Plan for Bullying in September 2023, which may address the challenges of fidelity and consistency across schools in the entire education system.

### 2.2. Safe Psychological and Physical School and Classroom Environments

The fundamental focus of a Whole Education Approach is on developing safe and caring cultures and contexts within schools. In order to manage school-based bullying, this perspective emphasises the creation of an environment where students and the whole school community feel safe, secure, welcomed, and supported. In order to achieve this safe environment, all school staff (e.g., teachers, Special Needs Assistants (SNAs), Special Needs teachers, guidance counsellors, administrative staff, bus drivers, and caretaking staff) are encouraged to participate in fostering a caring school environment that discourages bullying behaviour. The creation of a safe, caring, and inclusive school culture cannot be left to teachers, SNAs, or SENco staff in schools. This aligns with the whole school curricular modes for supporting pupil well-being, which emphasise the importance of creating a safe school.

It goes without saying that, in relation to inclusive education, the whole school policy and developing a culture of engagement with inclusion are also key policy objectives for inclusive education. The UNESCO "Global Education Monitoring Report 2020: Inclusion and education: All Means All" report [4] outlines a holistic vision for inclusion that is responsive to the diverse learning needs of all students and promotes their social, emotional, and academic development. This emphasises all school staff participating in creating a culture of inclusion that provides a well-rounded education that addresses the cognitive, emotional, social, and physical aspects of learning within a care-based school environment. However, such a model is heavily influenced by funding to support staff professional development, self-efficacy, and the development of positive attitudes towards working inclusively with diverse student populations [12,13,47].

### 2.3. Training and Support for School Staff

The UNESCO scientific committee strongly recommended that pre-service and in-service training on bullying prevention and intervention be provided to teachers and other school staff to support their engagement with the prevention of school-based bullying [2]. It has been identified that there is currently a gap in teacher education internationally, which contributes to a lack of engagement or efficacy among school staff in the management of

bullying as a phenomenon [8,46,48]. This point is also mirrored regarding the embedding of inclusive education within pre-service teacher education [18], with recent research finding pre-service teachers lack confidence in their knowledge and ability to implement inclusive practices in schools and a desire for more support in this area [40]. The Whole School vision is expressed within the UNESCO "Inclusion and Education: all means all" [4] report, which emphasises the importance of educational inclusion globally and the need to provide equitable and inclusive learning environments that accommodate diverse learning needs. However, to prepare teachers and school staff to enact such a system, providing appropriate pre-service and CPD for inclusive practice is imperative, and this needs to be rooted in the wider context within which student teachers live and work.

### 2.4. Curriculum, Learning, and Teaching to Promote a Caring School Climate

Like in the case of inclusive education, the UNESCO Scientific Committee makes the point that school climate is a key factor in managing relational social phenomena in school systems, such as bullying [2]. However, there are significant conceptual challenges in defining or operationalising the concept of school climate or culture [46]. A vital feature of the Whole Education Framework is the identification of the school as a caring and safe environment for all students, with teachers and other school staff acting as important models for developing such an environment. A caring and bullying-free school atmosphere can be achieved through the support of teachers and other staff members [49]. The use of interactive and engaging pedagogical and classroom practices is suggested in order to provide opportunities to model positive social interactions and increase school connectedness [8]. It was suggested that teachers should use a range of interactive strategies to engage students and develop their abilities in relation to decision-making and problem-solving, teamwork, and communication skills [50]. Essentially, the quality of interpersonal relationships and the type of teaching and learning in a school determine the extent to which a school is perceived as caring and children are facilitated to enhance their empathetic skills [50].

The development of inclusive whole school cultures is also a key long-term policy focus that focuses on fostering a whole school culture of inclusion for students with AEN and disabilities [36]. Developing positive attitudes towards inclusion across the spectrum of school staff, students, and other stakeholders is well represented within policy at both the national and international levels. However, inclusive education, in real terms, means changing the way things are normally done, which can be difficult when it is perceived as an 'inherently troubled and troubling educational and social project' [30] (p. 160) that means different things to different people. Key barriers to teacher engagement are a perceived lack of available time, inadequate classroom support from administrators, a mismatch between teacher style and inclusive pedagogy, and a lack of teacher understanding about factors that could assist the inclusion process.

### 2.5. Reporting Mechanisms with Support and Referral Services

The role of teachers and other staff as key actors in the monitoring of incidences or patterns of bullying within schools was identified by the UNESCO Scientific Committee [2] and a panel of experts on school-based bullying [3]. School staff being available and consistency in approach to monitoring bullying by staff were identified as critical factors in supporting students affected by bullying [51]. The need for a structured and clear approach led to the suggestion of having an anti-bullying coordinator in each school to act as a key organiser and point of contact [8]. Such a role is similar to the Special Needs Coordinator (SENco) role within schools, which supports the development of inclusive practices and ensures consistency of approach across whole school systems [52]. The need for a consistent point of contact with schools and an individual to advocate for educational inclusion across the different layers of management, year heads and class tutors, and other school staff is essential. Teachers are encouraged to engage in collaborative practices and reflection to develop as reflective practitioners and foster a coordinated and holistic approach to inclusive education [53,54].

In the case of both school-based bullying and inclusive practice, coordinators link with management and key school organisational structures, such as Student Support teams, Guidance Counsellors, and home-school liaison personnel. By actively working with colleagues, external therapeutic staff, school support staff, and parents, teachers can build strong relationships that contribute to a comprehensive educational experience for students. This approach aligns with the concept of "becoming an active professional" as outlined in the framework proposed by Florian and Spratt [54]. The development of such clear structures within complex school systems supports consistent and timely review and management of practices within schools. Such structures would support greater consistency of approach and efficient responses to concerns regarding either bullying or challenges to inclusive practices or engagement across the school community. A Whole Education Approach situates these processes within the existing social, policy, and funding frameworks within which schools operate.

### 2.6. Collaboration and Partnerships between the Education Sector and a Wide Range of Partners

Collaboration and partnerships are crucial in the context of inclusive education, in particular to address the lack of consensus, challenges, and inequalities present in current practices [55]. The importance of collaboration among education stakeholders in the inclusion and academic achievement of students with AEN has been highlighted in the literature [55–58]. However, as noted, the concept of inclusive education remains complex and diverse, with varying perspectives on its definition and implementation [13]. Some view special education as a form of inclusion, while others emphasise the need to change school structures to be more inclusive [25,28]. This diversity has led to inconsistent interpretations and practices within schools [18].

Multiple interconnected factors have been identified in the literature as essential for the development of inclusive practices. These factors include policies, funding, school organisation and leadership, school climate, classroom strategies, curriculum design, teacher training, and collaboration [59]. Despite the extensive body of research on inclusion, there is still a lack of consensus among education stakeholders regarding its values, potential benefits for all learners and teachers, implementation strategies, and the necessary systemic changes. This presents a significant challenge to effectively foster inclusion across various levels, such as policy development, teacher professional development, school culture, classroom practices, and engagement with families and communities [60].

### 2.7. Involvement of All Stakeholders in the School Community, Including Parents

Collaboration and partnerships are essential in addressing the challenges and inequalities present in inclusive education. This includes engaging teachers, support staff, parents, and the wider community in developing a shared understanding and commitment to inclusive practices. Policy initiatives have outlined complex models for inclusion, requiring changes in school practices and structures [20,38]. As such, policy development and reform need to consider the roles and rights of a spectrum of stakeholders who contribute to the educational careers and engagement of pupils. Such a perspective is especially important in supporting educational engagement and attainment for pupils with AEN, who particularly depend on the support of parents, a range of school staff, and staff working for organisations/agencies external to the school (such as health service staff and clinicians). The importance of such stakeholders is explicitly recognised within existing inclusive legislation such as the EPSEN Act [61], but collaboration between stakeholders is explicitly considered in detail with contemporary policy advice such as the Autism Good Practice guidelines [62] and the UNESCO Whole Education Approach framework.

The UNESCO Whole Education Approach provides a framework for considering the broader social context and resource policies needed to support inclusive education effectively. A Whole Education Framework goes beyond the school level and considers societal values, attitudes, and government policies [2]. This broader perspective acknowledges the importance of collaboration and partnerships in creating inclusive education environments.

It highlights the limitations of a sole whole-school perspective and emphasises the need for a comprehensive framework that involves all stakeholders. Research exploring the attitudes of teachers working with pupils with AEN in mainstream school settings has shown that support from external support services, clinicians, and resource organisations is important in supporting teacher self-efficacy and appropriate inclusive practices within schools [53]. Close links with parents, external organisations, and participation within communities of practice would also mediate school domestication of inclusive policy initiatives [39] and a lack of coherence regarding the implementation of inclusive practices in schools [32].

### 2.8. Student Empowerment and Participation

The concept of inclusion is grounded in the principles of social justice, human rights, and the belief that schools should cater to the needs of all students, embracing all forms of diversity [63]. The empowerment and support for the participation of students within the inclusive practices enacted within schools align with a right-based [1] understanding of inclusive practice and the empowerment of learner voice to influence provision in schools [64]. According to the United Nations Convention on the Rights of the Child (UNCRC) [1], children have the right to freely express their opinions and be actively involved in matters that affect their lives, including their education. The realisation of this right involves four essential components: creating spaces for children to express their views, facilitating their ability to share their perspectives, actively listening to their voices, and responding to their input [64].

As has been previously discussed, while policy internationally has been moving towards a greater focus on inclusive practice and provision, many students with AEN continue to be enrolled in separate specialised settings [13]. Loreman, Deppeler, and Harvey [65] contend that practices such as providing students with AEN-reduced timetables in mainstream education, placing them in segregated classrooms within mainstream schools, or offering substantially different study programmes in regular classrooms do not align with the principles of inclusion. However, in order to enact sustainable and coherent inclusive practices within schools, an accepted values-based framework needs to be accepted by all within the school and adapted for translation into action [29]. To promote and strengthen inclusive practices in schools through the use of learner voice, significant changes are necessary from all stakeholders involved. This entails a willingness on the part of adults to actively listen and be open to shifting their perspectives. It also involves engaging in meaningful conversations and dialogue with students to better understand their experiences and aspirations [66]. This process should be differentiated to support accessibility and access for all pupils across the enrolment, irrespective of profile or preferences [67], thus offering them an opportunity to express their perspectives [66].

### 2.9. Monitoring of Bullying and Evidence of Successful Responding

The development of clear and easily accessible approaches to reporting bullying incidents within schools is recognised within the Whole Education Approach [2]. In their engagement with a panel of experts on school-based bullying interventions, O'Higgins-Norman et al. [3] report that experts highlighted the importance of schools and educational authorities ensuring that staff were available and responsible for monitoring bullying in schools and that there were supports available to students who were affected by bullying. Research has identified that reporting mechanisms need to be seen as effective, or otherwise they will feed into a reluctance on the part of students to report bullying when it occurs [51]. Clear pathways for reporting were suggested to mitigate this challenge, and close links with the community and wider social or technological company stakeholders were also advised [3].

However, within the domain of inclusive education, the role of monitoring is slightly different. Monitoring has become important in recent years as international education policy has increasingly focused on the inclusion of all children within mainstream class settings, which in Ireland is linked to the government's obligations following the ratification of

the UNCRPD in 2018. The United Nations (UN) Committee that monitors implementation of the Convention has already advised that having a mainstream educational system and a separate special education system is not compatible with its view of inclusion and that parallel systems are not considered inclusive [41].

However, the implementation of such structural inclusive education changes faces challenges related to funding, policy, and social factors [18]. Monitoring the impact of inclusive education is crucial for accountability, identifying areas for improvement, and promoting evidence-based decisions. It allows for assessing progress, addressing gaps, and facilitating collaboration among stakeholders. Monitoring ensures continuous improvement in policies and practices to provide equitable and quality education for all students [4]. Without monitoring, it is difficult to assess whether the policy measures and practices are effective in promoting equitable and quality education for all students [18]. Monitoring enables evidence-based decision-making. By systematically collecting and analysing data on the impact of inclusive education policies and practices, policymakers can make informed decisions about resource allocation, policy adjustments, and professional development needs. Monitoring also helps identify gaps, challenges, and areas for improvement to ensure that resources are targeted where they are most needed and that efforts are focused on strategies that have proven to be effective in promoting inclusive education.

Within a Whole Education Approach, monitoring also facilitates the sharing of best practices and the dissemination of knowledge [2]. Through the collection and analysis of data on successful inclusive education initiatives, policymakers and educators can identify and showcase models of good practice that can be replicated in other settings. It provides an opportunity for parents, students, educators, and community members to be involved in the evaluation and assessment of inclusive education efforts. This sharing of knowledge and experience can contribute to the continuous improvement of inclusive education policies and practices at local, national, and international levels.

## 3. Conclusions

The Whole Education Approach proposed by the UNESCO Scientific Committee provides a useful lens for understanding and developing inclusive education. While inclusive education provision has seen increasing prominence across many European countries, seeking to encode social justice principles and adhere to international conventions, the reality within schools has been more complex [13,18,35]. The diversity of approaches to operationalizing inclusion within schools may reflect the ongoing lack of consensus on a definition or understanding of inclusive education [13]. Teacher attitudes and practices in relation to inclusion vary, and there are challenges in resource allocation and government commitment to inclusive practices [32,35]. The increasing number of students with additional educational needs and the (disproportionate) attendance of some groups in segregated provision raise concerns about accessibility and effectiveness [35].

Inclusive education policy and practice are at a crossroads, with a lack of clarity and representation of the holistic vision of inclusion [14]. In Ireland, there has been an increase in separate special classes for autistic pupils, operating as segregated provision despite national policies promoting inclusion [18,34]. This approach may be an administrative convenience, but there is a lack of policy linkage for long-term appropriate placement and social inclusion [35]. Schools tend to interpret policies to align with their existing preferences or systems [18,35,39]. This dynamic is perhaps understandable given a policy emphasis on inclusive practice in schools that requires changes to existing structures, which can be challenging for teachers and school leaders [36]. Collaborative agreement and participation from all stakeholders in the school community are needed to achieve coherent inclusion [32]. This foregrounds the utility of a holistic and integrated policy framework to inform the design, development, and comprehensive implementation of future initiatives for inclusive education. The Whole Education Approach provides just such a framework, identifying societal values, attitudes, and resource allocation as necessary for effective inclusive education [2]. It has been noted that segregated models of provision to support

pupils with AEN in mainstream schools and classes can often more closely resemble integration than inclusive practices, as the existing practices and modes of education remain largely unchanged while providing placement for these pupils [26]. This approach demonstrates a lack of integrated policy development and resourcing, which has been criticised for letting the mainstream system off the hook in relation to inclusive practices and for preventing the establishment of more sophisticated models of support [68]. Those in favour of more coherent and holistic systems of inclusion argue that integration, whether via the provision of special classes or inflexible practices within mainstream class settings, can amount to the segregation of pupils based on disability [68]. The logical conclusion of such perspectives is that such modes of education are discriminatory in nature and inconsistent with an inclusive education system.

The vision for inclusion in the UNESCO GEM "All means all" report [4] emphasises the importance of a holistic and high-quality education for all children and young people, promoting a sense of belonging and social participation [4]. This report assessed progress towards Sustainable Development Goal 4 (SDG 4) on education and focused on inclusion in education for children excluded because of background or ability [4]. It acknowledges the role played by the wider education system, local community attitudes, and society in creating an inclusive environment [3]. The approach aligns with the need to move away from segregated provision and towards inclusive practices within mainstream school settings to support long-term participation and inclusion for marginalised cohorts of children within society over their developmental trajectories [15,35]. While such a vision is laudable, the design and implementation of a holistically and socially inclusive education policy agenda is a complex task that requires a comprehensive framework to support an integrated systemic approach. The Whole Education Approach provides such a multi-element framework to guide coherent reform and policy development. It recognises that inclusive education goes beyond the responsibility of individual schools and requires a comprehensive framework that includes wider funding, education policy, and social factors. This perspective also aligns with existing international policy frameworks, such as the Salamanca statement [15] and the UNCRC [1] and UNCRPD [17].

Implementing the Whole Education Approach within inclusive education policy can support the more effective and holistic promotion of social inclusion, underpin equal opportunities, and recognise the diverse needs of all learners. The consideration of how societal values, policy, and resourcing intersect in understanding how interventions are operationalised is inherent within the Whole Education Framework, thus informing policy development from the outset [3]. In particular, the development of inclusive education requires collaboration and cohesion among school staff and wider social stakeholders within an inclusive education environment. Collaboration among stakeholders is promoted, including policymakers, educators, and communities, to create inclusive educational settings that foster a sense of belonging and prepare students for active participation in society. In addressing policy, funding, and wider social processes, the Whole Education Approach provides a deeper and more effective understanding of inclusive education, addressing the complexities and challenges involved in creating inclusive environments. Ultimately, this approach creates a real opportunity to achieve sustainable inclusive education provision and understand what it might take to achieve this.

**Author Contributions:** Conceptualization, All authors. writing—original draft preparation, N.K.; writing—review and editing, All authors. All authors have read and agreed to the published version of the manuscript.

**Funding:** This research received no external funding.

**Informed Consent Statement:** Not applicable.

**Data Availability Statement:** Not applicable.

**Conflicts of Interest:** The authors declare no conflict of interest.

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
