# Peer review of "A Whole Education Approach to Inclusive Education: An Integrated Model to Guide Planning, Policy, and Provision"

_education, doi:10.3390/educsci13090959_

Round 1

Reviewer 1 Report

Thank you for the chance to read this draft paper.

I found it well written with a clear line of argument. 

I think the general discussion around inclusive eduction is well handled. I then liked the specific focus on issues around bullying as an under-considered aspect to ensuring more inclusive practice. The use of the UNESCO framework is well handled and useful. 

However for me this focus doesn't seem well represented in your title for the article. The 2nd element of the title 'Conceptualising a Sustainable Future for Inclusive Education Provision'  is far broader than the focus of the piece and for me sets up different expectations. Perhaps aligning the title with your description in the intro ('relevance to the domain of inclusive education policy and provision') in someway might be clear for readers?

I also think the combination of 2 section titles and numbers (2.6. Collaboration and partnerships between the education sector and a wide range of partners, & 2.7. Involvement of all stakeholders in the school community, including parents)  needs re-thinking. 

I'm guessing you had to reduce the length and merge sections? 

For a stand alone piece this is confusing, so I suggest a single sub-section header be used and the subsequent sections re-numbered. 

I hope these comments are useful for finalising the article ready for publication.

Author Response

Dear Reviewer, 

Many thanks for your review of our manuscript and your comments and suggestions. We have revised our manuscript in light of your suggestions and outline the changes we have made in the below table. We have also attached a revised draft of our manuscript with the changes highlighted in yellow, for your consideration. 

Many thanks again. 

Reviewer 1: However for me this focus doesn't seem well represented in your title for the article. The 2nd element of the title 'Conceptualising a Sustainable Future for Inclusive Education Provision'  is far broader than the focus of the piece and for me sets up different expectations. Perhaps aligning the title with your description in the intro ('relevance to the domain of inclusive education policy and provision') in someway might be clear for readers?

An alternative heading has been suggested which is highlighted in yellow.

Reviewer 1: I also think the combination of 2 section titles and numbers (2.6. Collaboration and partnerships between the education sector and a wide range of partners, & 2.7. Involvement of all stakeholders in the school community, including parents)  needs re-thinking.

For a stand alone piece, this is confusing, so I suggest a single sub-section header be used and the subsequent sections re-numbered.

These sections have been separated and each have been augmented to support coherence. Changes have been highlighted in yellow.

Reviewer 2 Report

The authors offer theoretical research and raise issues of inclusive education, but, in my opinion, one of the most important aspects is that “values-based framework needs to be accepted by all within the school” (p.10). It would be interesting to see an in-depth study of why in Ireland, there has been an increase in separate special classes for autistic pupils…

The aim of the paper “…to discuss the relevance of recent developments in the field of school-based bullying intervention and assess their applicability to the policy discourse of inclusive education” generally achieved.

I think the paper is important in terms of raising the issue, but unfortunately the conclusions are not innovative. One of the conclusions “The Whole Education Approach recognizes that inclusive education goes beyond the responsibility of individual schools and requires a comprehensive framework that includes wider funding, education policy, and social factors”… Hasn't this been known for a long time?

The research methodology is not described.

Author Response

Dear Reviewer, 

Many thanks for your time in reviewing our manuscript and for your helpful comments. We have revised our manuscript in response to your suggestions. Please see the below table of changes we have completed in response to your comments. We have also attached a revised draft of our manuscript which has the revisions highlighted in yellow for your consideration. 

Many thanks again. 

Reviewer 2: I think the paper is important in terms of raising the issue, but unfortunately the conclusions are not innovative.

The conclusions have been revised to support clearer messages and to address the utility of the WEA in supporting clear, sustainable policy development which builds collaboration and participation into to inclusive practice from the outset. Changes have been highlighted in yellow.

Reviewer 2:. “values-based framework needs to be accepted by all within the school” (p.10). It would be interesting to see an in-depth study of why in Ireland, there has been an increase in separate special classes for autistic pupils…

Such a detailed exploration of use of separate specialist classes in Ireland is beyond the scope of the current article. It has, however been discussed in great detail in Travers (2023) which is referenced in the conclusion of the current study. That article is published within the same special issue that the current manuscript has been submitted to also.

Round 2

Reviewer 2 Report

Thank you for your research